# Dynamics of Multipole Solitons and Vortex Solitons in PT-Symmetric Triangular Lattices with Nonlocal Nonlinearity

**Jing Huang [1,2], Yuanhang Weng [1], Peijun Chen [1] and Hong Wang [1,3,\*]**

[1]   Engineering Research Center for Optoelectronics of Guangdong Province,
      School of Electronics and Information Engineering, South China University of Technology,
      Guangzhou 510640, China
[2]   School of Materials Science and Engineering, Guizhou Minzu University, Guiyang 550025, China
[3]   Zhongshan Institute of Modern Industrial Technology, South China University of Technology,
      Zhongshan 528437, China
\*   Correspondence: phhwang@scut.edu.cn



**Featured Application:** The well-defined shape and robustness of spatial optical solitons in nonlocal nonlinear and parity–time (PT)-symmetric triangular lattices provide a feasible scheme for creating reconfigurable all-optical circuits. Specially, multipole solitons and vortex solitons that have complex phases and wave fronts may improve the ability to carry information significantly.

**Abstract:** We investigate the evolution dynamics of solitons with complex structures in the PT-symmetric triangular lattices with nonlocal nonlinearity, including dipole solitons, six-pole solitons, and vortex solitons. Dipole solitons can be linearly stable with a small degree of gain/loss, while six-pole solitons can only be stable when both the degree of gain/loss and the degree of nonlocality are small. For unstable solitons, some humps will decay quickly or new hotspots will appear during propagation. According to the existence range of dipole solitons, the multipole solitons tend to exist in PT-symmetric triangular lattices whose nonlocal nonlinearity is intermediate. We also consider the vortex solitons with high topological charges in the same triangular lattices and find that their profiles are codetermined by the propagation constant, degree of nonlocality, and topological charge.

**Keywords:** PT symmetry; nonlocal nonlinearity; triangular lattices; multipole solitons; vortex solitons

## 1. Introduction

Optical modes tend to diffract after transmitting in a waveguide, which is detrimental to the long-haul transmission of optical signals. If the diffraction is balanced by a self-focusing effect, which is the result of the inherent nonlinearity of some media, solitons are formed and maintain their profiles and power over a relatively long distance. Nonlinearities could be local or nonlocal. In the nonlocal case, the nonlinear response depends on the light incidence area and the neighborhood, while the local nonlinear response depends only on the single incidence area. Practically, nonlocal nonlinearity is demonstrated in various materials, including nematic liquid crystals [1–3], thermal nonlinear media [4–6], and so on. Nonlocal nonlinearity strongly affects the propagation of light, providing the potential to form and stabilize many types of solitons, such as defect solitons [7], breather solitons [8], surface solitons [9], multipole solitons [10], and vortex solitons [11].

On the other hand, parity–time (PT) symmetry has drawn much attention since it was first proposed by Bender and Boettcher [12]. A PT-symmetric system is required to satisfy the PT transformation, which is defined by the parity operator **P** as $\mathbf{P}\psi(\mathbf{r}, t) = \psi(-\mathbf{r}, t)$ and the time reversal operator **T** as $\mathbf{T}\psi(\mathbf{r}, t) = \psi^*(\mathbf{r}, -t)$, where $\psi(\mathbf{r}, t)$ is a wave function in quantum mechanics. A PT-symmetric system is proved to have entirely real spectra even if its Hamiltonian is non-Hermitian [12,13]. In optics, the refractive index of a two-dimensional (2D) PT-symmetric system is required to be complex and meet $n(x, y) = n^*(-x, -y)$; that is, its real part is an even function, and the imaginary part is an odd function. From a practical point of view, the imaginary part of the refractive index is realized by inducing optical gain and loss [14,15]. By means of a PT-symmetric refractive index, a nonlinear medium with complex potential can be fabricated to support complicated solitons, such as multipole solitons [16–18] and vortex solitons [19–21].

Much research has focused on solitons in square nonlinear optical lattices, which can be induced by interfering two pairs of ordinarily plane-polarized light [22]. In contrast to square lattices, triangular lattices are more compact and they support larger bandgaps [23]. Thus, triangular lattices are expected to exhibit new features of solitons upon propagation, such as the periodic beam oscillation inside a waveguide [24], and a certain parameter range in which solitons can exist or be stabilized [25,26]. However, most of the previous works on solitons in triangular lattices are about local solitons. The evolution dynamics during the propagation of solitons in nonlocal triangular lattices with PT symmetry are still unexplored. In practice, the dynamics and stability of spatial optical solitons and the interaction between them can be used in optical interconnects [27,28], beam manipulation [29], optical computing [30], all-optical switching [31,32], and logic gates [33–35]. With appropriate improvements, solitons in nonlocal nonlinear and PT-symmetric triangular lattices may achieve the same function, thus providing a scheme for creating reconfigurable all-optical circuits.

In this paper, we investigate in detail the propagation and stability of the multipole solitons, as well as the evolution of the vortex solitons in a nonlocal triangular optical lattice with PT-symmetric complex potentials. These two types of soliton have complex phases and wave fronts, which means that they will have more complex dynamics and carry more information than fundamental solitons. We find that the dipole and the six-pole solitons can be found in the first bandgap of the specific triangular lattices. According to the numerical calculation, their stability is demonstrated to be strongly influenced by the degrees of nonlocalities and the gain/loss coefficient. As for the vortex solitons, they exist in both the first bandgap and the infinite bandgap of the triangular lattices. The profiles of these vortex solitons are influenced by the propagation constant, the degree of nonlocality, and the topological charge.

## 2. Theoretical Model

The model of the nonlocal nonlinear medium with complex potentials can be established by the normalized 2D coupled nonlinear Schrödinger equation as follows [24,26,36], with the transverse coordinates $x$ and $y$ scaled to the input beam width, $w_0$, and the longitudinal coordinate $z$ scaled to the diffraction length, $L_0 = 2kw_0^2$:

$$\begin{cases} iU_z + \nabla_\perp U + (V + iW)U + \sigma nU = 0 \\ d\nabla_\perp n - n + |U|^2 = 0 \end{cases} \tag{1}$$

where $\nabla_\perp = \partial_{xx} + \partial_{yy}$ indicates the 2D Laplace operator. This model can be realized in the liquid crystal E7 [37]. The dimensionless light field amplitudes $U_n$ and the nonlinear refractive index $n$ are scaled to $2E\sqrt{\varepsilon_0\Delta\varepsilon_E}/\sqrt{\pi}\Delta\varepsilon_L k_0 w_0$ and $\varepsilon_0/k_0^2 w_0^2 \Delta\varepsilon_E$, respectively, where $k_0$ is the wave numbers in a vacuum, $\varepsilon_0$ is the vacuum permittivity, and $E$ is the electric field strength. The material parameters of liquid crystals, namely $\Delta\varepsilon_E$, $\Delta\varepsilon_L$, and $K$, correspond to the anisotropy at the frequency of the quasistatic electric field, the anisotropy at the light frequency, and the elastic constant, respectively. $d$ is the degree of nonlocality, defined as $d^2 = \pi K/2w_0^2 E\Delta\varepsilon_E$. $\sigma = \pm 1$ represents the focusing or defocusing

nonlinearity. Here, we fix $\sigma = -1$, which corresponds to the defocusing type. $(V + iW)$ represents the PT-symmetric complex potential. $V$ and $W$ are the real part and the imaginary part, respectively. To simulate the triangular lattices, we set the PT complex as:

$$\begin{cases} V = 2V_0\left[\frac{3}{2} + \cos\left(k_0 x + \frac{k_0}{\sqrt{3}}y\right) + \cos\left(k_0 x - \frac{k_0}{\sqrt{3}}y\right) + \cos\left(\frac{2k_0}{\sqrt{3}}y\right)\right] \\ W = \frac{4}{3}W_0\left[\sin\left(k_0 x + \frac{k_0}{\sqrt{3}}y\right) + \sin\left(k_0 x - \frac{k_0}{\sqrt{3}}y\right) + \sin\left(\frac{2k_0}{\sqrt{3}}y\right)\right] \end{cases} \tag{2}$$

where $V_0$ is the depth of the PT-symmetric lattices and $W_0$ is the gain/loss coefficient. These triangular lattices can be fabricated by interfering three ordinarily polarized broad beams [36]. No linear mode can exit via the bandgaps, but gap solitons exist there. Thus, we mainly concentrate on the nonlinear propagation modes in bandgaps. In addition, the threshold of PT symmetry breaking is found to be $W_{0th} = 3$, at which the first bandgap and the infinite bandgap merge together.

We substitute $U(x, y, z) = q(x, y)e^{i\mu z}$ in Equation (1), and thus it becomes:

$$\begin{cases} -\mu q + \nabla_\perp q + (V + iW)q + \sigma nq = 0 \\ d\nabla_\perp n - n + |q|^2 = 0 \end{cases} \tag{3}$$

where $q(x, y)$ is a complex function and $\mu$ is the propagation constant. Equation (3) can be numerically solved by the developed modified squared-operator method (MSOM). [38] The power of solitons is defined as $P = \int \int_{-\infty}^{+\infty} |q|^2 dx dy$.

We utilize the symmetric split-step Fourier method to simulate the propagation of solitons in the nonlinear triangular lattices. As for the linear stability, we add a small perturbation into the solitary solution:

$$U(x, y, z) = \left[q(x, y) + F(x, y)e^{\lambda z} + G^*(x, y)e^{\lambda^* z}\right] \tag{4}$$

where $|F|, |G| \ll 1$ is a small perturbation, $\lambda$ is the complex instability growth rate, and the superscript "*" represents the complex conjugation.

By substituting Equation (4) into Equation (1) and reducing the equations, we obtain the eigenvalue equations:

$$\begin{cases} \lambda F = i[-\mu F + \nabla_\perp F + (V + iW)F + \sigma nG + \sigma q\Delta n] \\ \lambda G = i[\mu G - \nabla_\perp G - (V - iW)G - \sigma nF - \sigma q^*\Delta n] \end{cases} \tag{5}$$

where $n(I) = \int_{-\infty}^{+\infty} \int_{-\infty}^{+\infty} R(x - x\prime, y - y\prime) \cdot I(x\prime, y\prime)dx\prime dy\prime$ depends on the 2D nonlocal response function $R(x, y) = \frac{1}{2\pi \sqrt{d}} \exp(-\sqrt{\frac{x^2+y^2}{d^2}})$ and the intensity $I(x, y) = |U(x, y)|^2$. Thus, the variation of $n$ is $\Delta n = \int \int_{-\infty}^{+\infty} R(x - x\prime, y - y\prime)[q(x\prime, y\prime)G(x\prime, y\prime) + q^*(x\prime, y\prime)F(x\prime, y\prime)]dx\prime dy\prime$. By calculating the eigenvalue $\lambda$, the linear stability analysis is as follows. If the real part of $\lambda$ is zero, the soliton is linearly stable; otherwise, it is linearly unstable. The value of the real part of $\lambda$ indicates the degree of linear instability, and thus $\lambda$ is named the perturbation growth rate.

To intuitively show the influences of different degrees of nonlocalities, we present three profiles of the nonlocal responses with $d = 0.05$, $d = 0.5$, and $d = 5$ corresponding to the weak, intermediate, and strong nonlocalities, respectively. As shown in Figure 1, the response of nonlinearity is nearly local when $d = 0.05$, and the nonlinearity affects the incident point and the area close to the point. When $d = 5$, the nonlinearity affects a wide area around the incident point, which is strongly nonlocal.

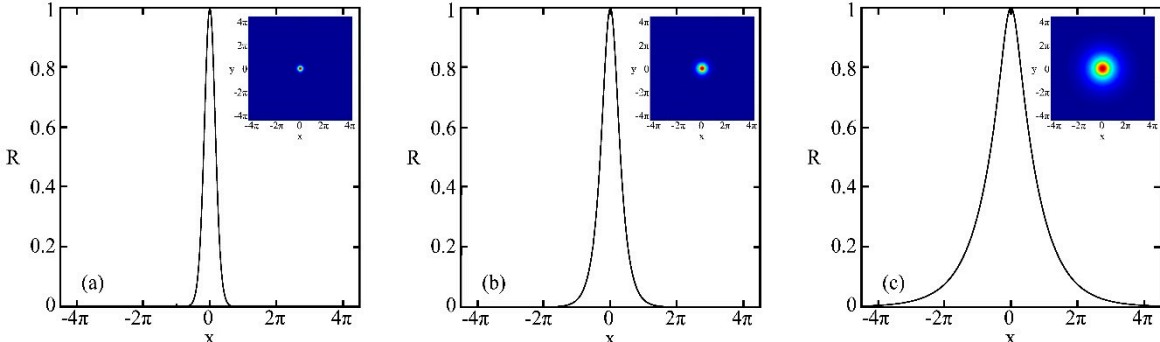

**Figure 1.** The response of nonlinearity, *R*, when (**a**) the degree of nonlocality (*d*) = 0.05, (**b**) *d* = 0.5, and (**c**) *d* = 5. In these figures, x refers to the horizontal coordinates. The insets of each picture are the corresponding transverse profiles of *R*. The amplitude of each *R* is normalized.

## 3. Numerical Results

### 3.1. Dipole Solitons

We first consider dipole solitons, because they are the basic type of multipole solitons. We find that dipole solitons in PT-symmetric triangular lattices can be generated with different degrees of nonlocality, *d*, and with different PT-symmetric potentials. As for PT-symmetric potentials, the primary parameter that we focus on is the degree of gain/loss, $W_0$. The power curves of dipole solitons with different *d* and $W_0$ are shown in Figure 2. It is intuitive that the nonlinear optical modes are bifurcated from the optical energy band. From the $\mu$–$W_0$ plane, where $\mu$ is the propagation constant, it is clear that the range of the first bandgap (i.e., the white area on the $\mu$–$W_0$ plane) becomes narrower as $W_0$ increases. Further, the first bandgap will disappear when $W_0 = 3$, which means the PT symmetry is broken. With the same $W_0$, dipole solitons can exist in weak nonlocality (*d* = 0.05), intermediate nonlocality (*d* = 0.5), and strong nonlocality (*d* = 2). As the nonlocality becomes stronger, the power of dipole solitons is greater, which corresponds to a steeper slope as seen in Figure 2. However, fewer solitons can be found when $W_0$ becomes greater. Generally speaking, stable dipole solitons tend to exist with weak nonlocality and small $W_0$. In the following discussion, we choose the PT-symmetric triangular lattices with $V_0 = 2$ and $W_0 = 0.3$, of which the first bandgap is $4.82 < \mu < 10.67$ and the infinite bandgap is $\mu > 10.75$.

To investigate in detail the dynamics of the dipole solitons, we simulate their propagation. For the stable dipole solitons, they can maintain their profiles and power over a relatively long distance (*z* > 100). We take the one with *d* = 0.5, $\mu = 9.6$ as an example, as presented in Figure 3. This dipole soliton keeps its profile until *z* = 140, as shown in Figure 3a,b. Its phase structure is plotted in Figure 3c, which indicates that the two humps of this soliton are in-phase. We also calculate the change of the gravity center, which is defined as $\Delta = \pm \sqrt{(x\prime - x_0)^2 + (y\prime - y_0)^2}$, where ± represents whether $(x\prime, y\prime)$ is on the right (+) or left (−) of the initial position $(x_0, y_0)$. The gravity center suffers a slight oscillation before it is dispersed over the whole space, which is shown in Figure 3d. Physically, the oscillation of the gravity center is due to the redistribution of transverse power. Similar to the gravity center, the power of the dipole soliton also undergoes an oscillation. Once this soliton cannot maintain its profile, the nonlinearity of the medium will lead to a catastrophic focusing of the light, and then the power increases rapidly, which may destroy the medium, as shown in Figure 3e. The linear stability spectrum is shown in Figure 3f, and the real part of the growth rate $\lambda$ is zero, which means that this dipole soliton is linearly stable against a small perturbation.

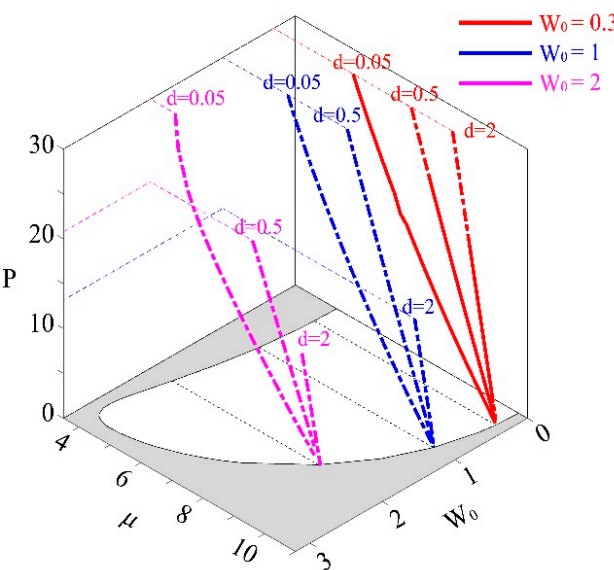

**Figure 2.** The power curves of dipole solitons in parity–time (PT)-symmetric triangular lattices with different nonlocality, *d*, and degree of gain/loss, $W_0$. The gray area in the $\mu$–$W_0$ plane (propagation constant $\mu$ versus degree of gain/loss $W_0$) is the optical energy band of the triangular lattices, thus the white area is the first bandgap. The solid lines indicate that the dipole solitons are stable, while the dashed lines indicate that they are unstable.

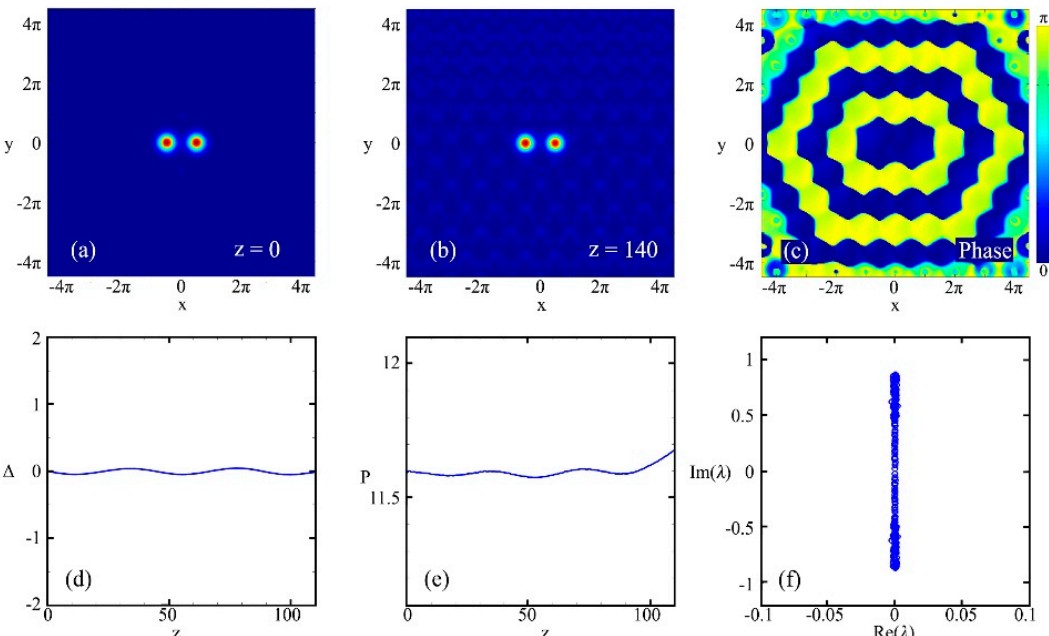

**Figure 3.** The evolution dynamics of stable dipole soliton with $d = 0.5$, $\mu = 9.6$. (**a**,**b**) correspond to the transverse profile of solitons at $z = 0$ and $z = 140$, respectively. (**c**) is the phase structure of the dipole soliton. (**d**,**e**) correspond to the change of the gravity center, $\Delta$, and the power of the soliton, *P*, during propagation. (**f**) is the spectrum of the perturbation growth rate, where $\text{Im}(\lambda)$ and $\text{Re}(\lambda)$ are the imaginary part and the real part of the instability growth rate $\lambda$.

As for the unstable dipole solitons, we consider the dipole soliton with $d = 0.5$, $\mu = 6$, as shown in Figure 4. The profiles of this soliton at $z = 0, 40, 150$ are plotted in Figure 4a–c, respectively. One hump of this dipole soliton decays quickly (before $z = 40$), and another hump can maintain its profile over a long distance (*z*). From Figure 4d, it is obvious that the gravity center shifts from $(0, 0)$ to $(-\pi, 0)$, which

means the right hump whose original position is $(\pi, 0)$ decays at the beginning of propagation and the power finally concentrates in the left hump. However, the power of this dipole soliton fluctuates violently during propagation, as shown in Figure 4e, which indicates this soliton is strongly unstable. The spectrum of $\lambda$ is plotted in Figure 4f, demonstrating the linear instability of this dipole soliton. The maximum of the real part of $\lambda$ is $\max[\mathrm{Re}(\lambda)] = 0.0929$.

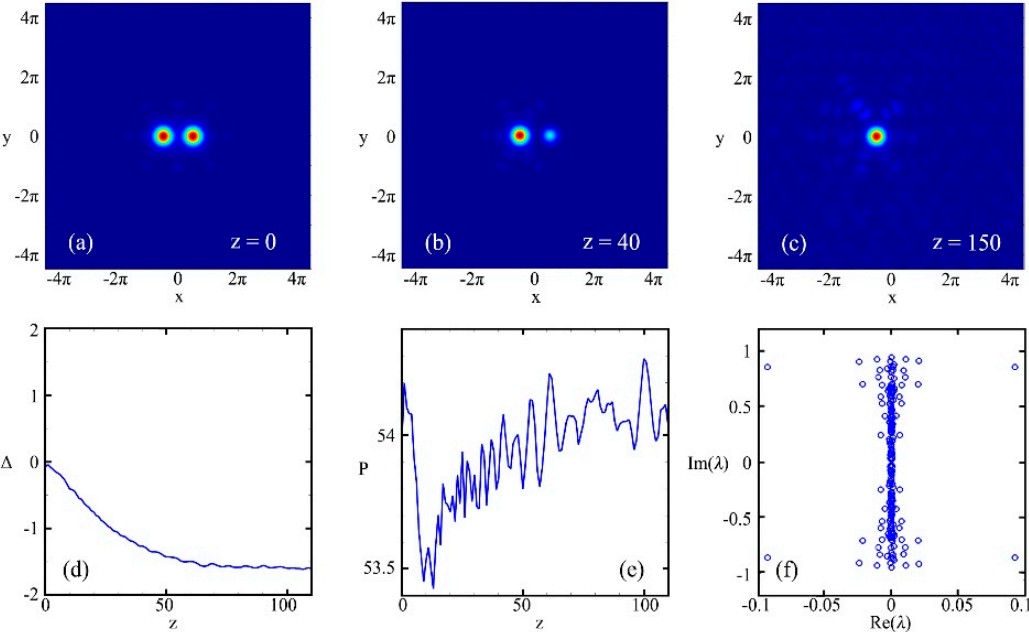

**Figure 4.** The dynamics of the unstable dipole soliton with $d = 0.5, \mu = 6$. (**a–c**) correspond to the transverse profile of solitons at $z = 0$, $z = 40$, and $z = 150$, respectively. (**d,e**) correspond to $\Delta$ and $P$ during propagation, respectively. (**f**) shows the spectrum of the perturbation growth rate.

When the nonlocality becomes stronger, dipole solitons with smaller $\mu$ will be unstable, as shown by the red lines in Figure 1. We investigate the unstable dipole soliton with $d = 2, \mu = 7.6$ (the first row in Figure 5) and the stable one with $d = 2, \mu = 10$ (the second row in Figure 5). The evolution dynamics of these solitons are the same as that of the intermediate nonlocal type. For the unstable soliton, the right hump decays before $z = 40$, and its linear stability spectrum indicates it is linearly unstable. The corresponding maximum of $\mathrm{Re}(\lambda)$ is $\max[\mathrm{Re}(\lambda)] = 0.0628$. For the stable soliton, the profile and power can be maintained before $z = 150$. Its gravity center suffers a slight oscillation, but the period of this oscillation is larger than the soliton with $d = 0.5, \mu = 9.6$. Further, the real part of $\lambda$ is zero, which indicates this soliton is stable.

In general, fewer multipole solitons can exist in the first gap with a larger degree of gain/loss, $W_0$, as Figure 2 shows. To investigate the influence of $W_0$ on the existence of multipole solitons, we calculate the maximum and minimum of propagation constants, $\mu$, under which dipole solitons can exist with different $W_0$. Three types of nonlocalities are considered separately, as shown in Figure 6. The green areas represent that dipole solitons can be found if their parameters are in these areas, and the shaded areas represent the optical energy band for linear optical modes where solitons cannot be found. With the weak nonlocality ($d = 0.05$), dipole solitons exist in almost the whole first bandgap, as shown in Figure 6a. As the nonlocality becomes stronger, the existence range becomes narrower. With strong nonlocality ($d = 2$), dipole solitons can only be found near the edge of the upper optical energy band, as shown in Figure 6c.

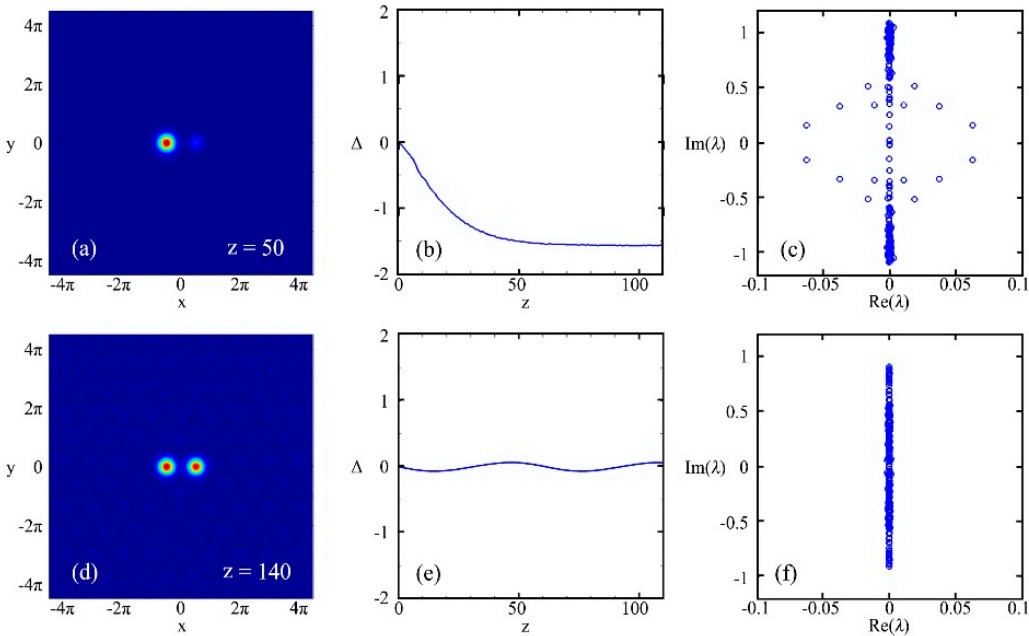

**Figure 5.** The evolution dynamics of strongly nonlocal dipole solitons. The first row corresponds to the unstable soliton with $d = 2, \mu = 7.6$, and the second row corresponds to the stable soliton with $d = 2, \mu = 10$. (**a,d**) show their transverse profiles. (**b,e**) show the changes of the gravity center, $\Delta$, during propagation. (**c,f**) show the spectrums of their perturbation growth rates.

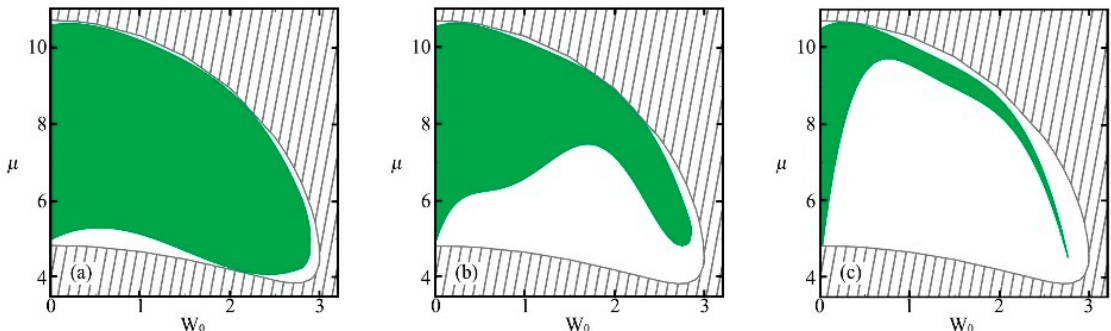

**Figure 6.** The existence range of dipole solitons in nonlocal nonlinear triangular lattices with different degrees of gain/loss. (**a–c**) correspond to weak ($d = 0.05$), intermediate ($d = 0.5$), and strong ($d = 2$) nonlocalities, respectively. The shaded areas represent the optical energy band for linear optical modes. The green areas represent the parameter ranges in which dipole solitons can exist.

### 3.2. Six-Pole Solitons

Another type of multipole solitons that we investigate is the six-pole solitons in the same PT-symmetric triangular lattices with nonlocal nonlinearity. Six-pole solitons with different collocations of $d$ and $W_0$ are found to exist, of which the power curves are plotted in Figure 7. The trends of these curves are as the same as the power curves of dipole solitons. However, we cannot find stable solitons with strong nonlocality ($d = 2$), which is presented as the red dashed line in Figure 7. The power of six-pole solitons is about 3 times as large as the power of dipole solitons, because of the difference in the number of humps. For $W_0 = 0.3$, the six-pole solitons with weak ($d = 0.05$) or intermediate ($d = 0.5$) nonlocality are stable enough to propagate without dispersion at $z > 140$. Moreover, all the power curves are bifurcated from the linear modes.

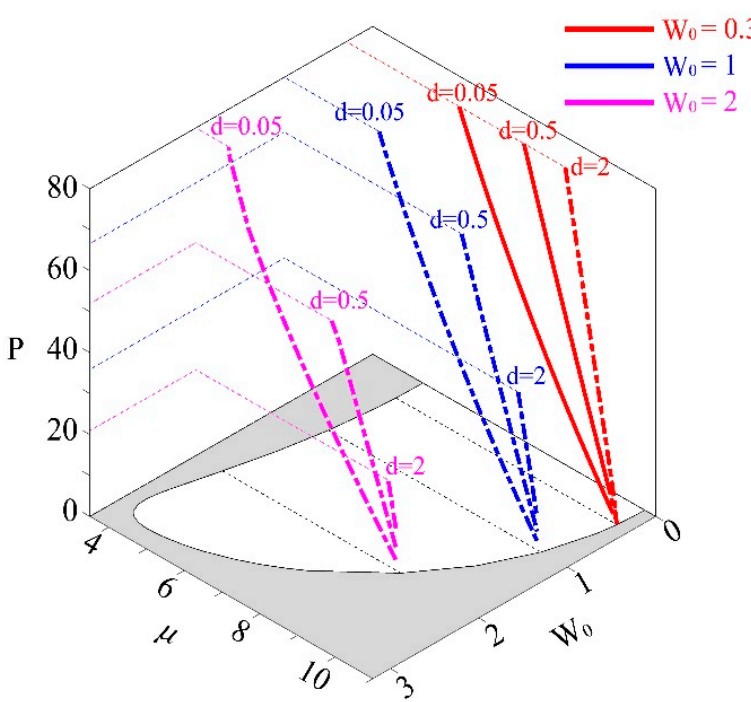

**Figure 7.** The power curves of six-pole solitons in PT-symmetric triangular lattices with different $d$ and $W_0$. The gray area in the $\mu$–$W_0$ plane is the optical energy band of the triangular lattices, and the white area is the first bandgap. The solid lines correspond to the stable solitons, while the dashed lines correspond to the unstable ones.

The six-pole soliton with $d = 0.5, \mu = 10$ is stable enough to propagate in the PT-symmetric triangular lattices. The profiles of this soliton at $z = 0$ and $z = 140$ are shown in Figure 8a,b, the structures of which are regular hexagons. The phase structure is plotted in Figure 8c, indicating its six humps are in-phase. We simulate the propagation of this soliton and calculate its gravity center and power in every step. The results are shown in Figure 8d,e. The transverse power distribution of this soliton is quite uniform, and thus the gravity center is almost at its original position. However, this six-pole soliton cannot maintain its profile after $z = 140$. This is because the nonlinearity will cause the oscillation of soliton power and increase the power catastrophically at a long distance. The result of linear stability analysis is plotted in Figure 8f. $\mathrm{Re}(\lambda)$ is zero, demonstrating this six-pole soliton is linearly stable.

In contrast to the intermediate nonlocality, the dynamics of six-pole solitons in the PT-symmetric triangular lattices with strong nonlocality are more complicated. These solitons may be concentrated into some specific points or create new hotspots. We present two unstable six-pole solitons with $d = 2, \mu = 8.4$ (the first row in Figure 9) and $d = 2, \mu = 10.4$ (the second row in Figure 9). For the six-pole soliton with $d = 2, \mu = 8.4$, some humps of this soliton decay quickly, as shown in Figure 9a. Then, its power will shift into other humps, which increases the power density in the medium and may burn the medium. Figure 9b,c shows that the power of this soliton suffers a strong oscillation and eventually is concentrated into the left two points. The spectrum of the perturbation growth rate is plotted in Figure 9d, and $\max[\mathrm{Re}(\lambda)] = 0.0146$. For the six-pole soliton with $d = 2, \mu = 10.4$, its initial profile is a hexagon with a weak hotspot in its center, as shown in Figure 9e. However, the power of that hotspot becomes stronger during propagation, which makes the soliton turn into seven humps, as shown in Figure 9f. From the evolution of power, which is plotted in Figure 9g, the power of this soliton is maintained until $z = 130$. Thus, the six humps of the soliton and the hotspot together average the initial power. Although the "7-pole" soliton can maintain its profile and power over a relatively long distance, the perturbation growth rate that we calculate is $\max[\mathrm{Re}(\lambda)] = 0.0043$, which means this soliton has minor linear instability, as shown in Figure 9h. The physical mechanisms of these

two different transverse instabilities are as follows. The strong nonlocality broadens the refractive index distribution of each soliton hump, and those broadened areas mainly overlap on the center of the six-pole soliton, which creates a hotspot of the refractive index. Then, the soliton energy "flows" into this hotspot and enhances the overlap effect. Eventually, the power of this hotspot is equal to the other humps of the soliton.

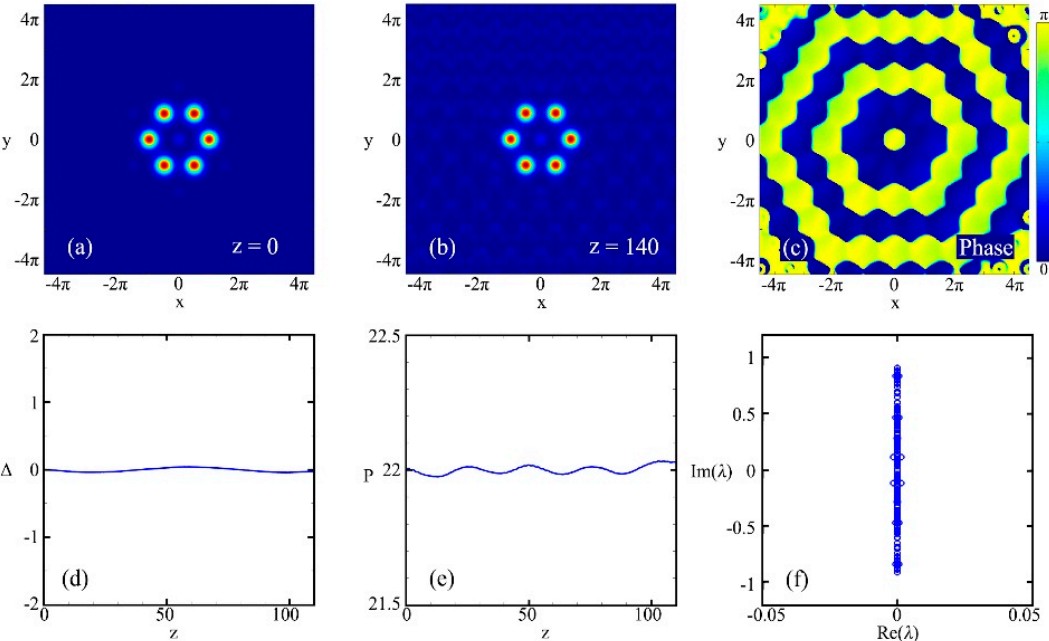

**Figure 8.** The evolution dynamics of stable six-pole solitons with $d = 0.5, \mu = 10$. (**a,b**) correspond to the transverse profile of the solitons at $z = 0$ and $z = 140$, respectively. (**c**) shows the phase structure of the six-pole soliton. (**d,e**) correspond to $\Delta$ and $P$ during propagation. (**f**) shows the spectrum of the perturbation growth rates.

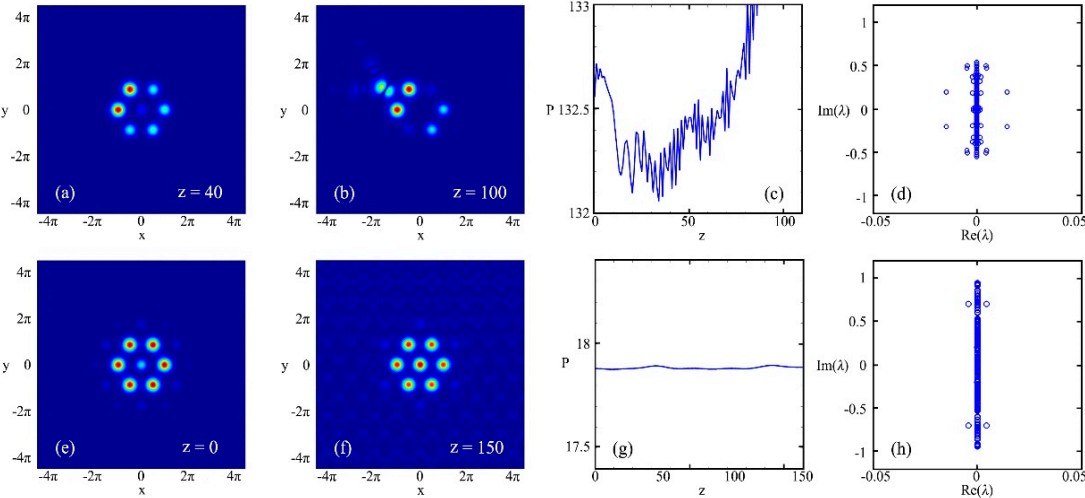

**Figure 9.** The dynamics of strongly nonlocal six-pole solitons. The first row corresponds to the unstable soliton with $d = 2, \mu = 8.4$, and the second row corresponds to the stable soliton with $d = 2, \mu = 10.4$. (**a,b,e,f**) show their transverse profiles. (**c,g**) show the power of the soliton, $P$, during propagation. (**d,h**) show the spectrums of their perturbation growth rates.

It is worth emphasizing that the existence range of six-pole solitons is similar to the existence range of dipole solitons in Figure 6, except that the existence range of six-pole solitons becomes narrower more quickly when nonlocality becomes stronger.

### 3.3. Vortex Solitons

Unlike multipole solitons, vortex solitons have more complicated phase structures and carry angular momenta. The main difference between the multipole solitons and vortex solitons is their phase structures. For the multipole solitons that are discussed above, their phase structures are in-phase. For vortex solitons, their phase structures depend on their topological charges, *m*. Moreover, the sign of *m* denotes the direction of the angular momentum (counterclockwise for $m > 0$, clockwise for $m < 0$).

We find that the PT-symmetric triangular lattices with nonlocal nonlinearity can support vortex solitons with different topological charges. To compare the phase structures of the vortex solitons, we choose three solitons with $m = 1, 2, 3$, as shown in Figure 10. Their other parameters are the same, i.e., $d = 0.5, \mu = 10$. They have the same profile as Figure 10a, which is composed of six humps. All their phase structures have gradual phase differences in a spiral form. In Figure 10b,d, we show the phase structures of these vortex solitons. For the vortex soliton with unity topological charge ($m = 1$), its phase structure is restored to its original shape after rotating by $2\pi$. For a vortex soliton $m = 2$, the phase structure is restored to its original shape after $\pi$ of rotation. As for the one with $m = 3$, the angle of phase structure recovery is $2\pi/3$. The circles in Figure 10 denote the positions of the soliton humps. During the propagation of these vortex solitons, their phase structures rotate counterclockwise. We take the vortex solitons with $m = 3, d = 0.5, \mu = 9$ as an example, which is shown in Figure 11.

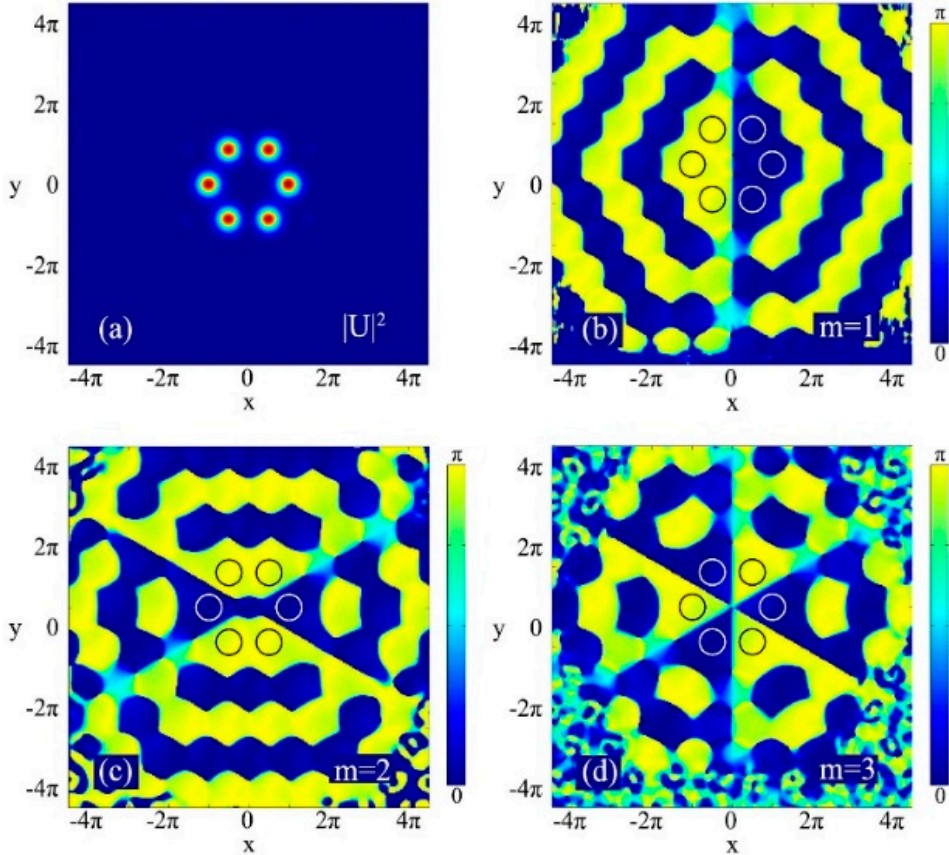

**Figure 10.** The profile and the phase structures of vortex solitons with different topological charges. (**a**) is their profile $|U|^2$, i.e., square of modules of soliton solution $U$. (**b**–**d**) correspond to the phase structures of vortex solitons with $m = 1, 2, 3$. $m$ = topological charges.

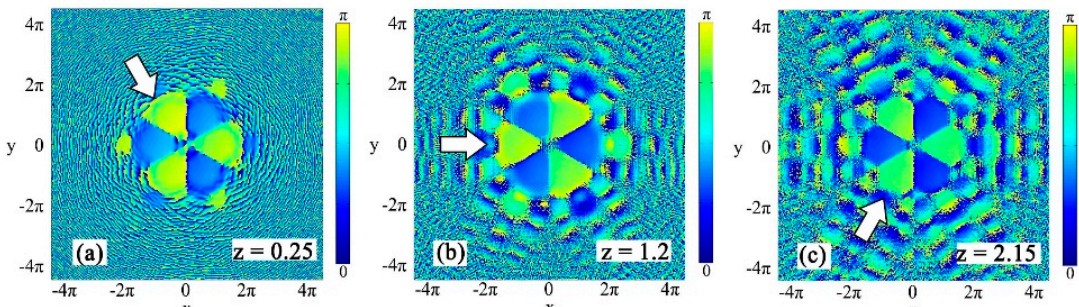

**Figure 11.** The phase structure evolution of vortex solitons with $m = 3, d = 0.5, \mu = 9$. (**a–c**) correspond to the phase structures of vortex solitons with $z = 0.25, 1.2, 2.15$. The arrows point to the phases of one specific peak.

According to numerical calculations, the vortex solitons can exist in both the first bandgap ($4.82 < \mu < 10.67$) and the infinite bandgap ($\mu > 10.75$). For the multipole solitons discussed above, the stable solitons have the same profiles (dipole or six-pole). For vortex solitons, however, their profiles are influenced by the propagation constant, $\mu$, the degree of nonlocality, $d$, and the topological charges, $m$. These vortex solitons can maintain their initial profiles during the propagation past the point where $z = 140$. To illustrate this novel property, we take vortex solitons with $m = 2$ or $m = 3$ and with different $\mu$ as an example. The profiles of the vortex solitons under intermediate nonlocal nonlinearity ($d = 0.5$) with $m = 3$ are shown in Figure 12. The upper row corresponds to the vortex solitons in the first bandgap, and the lower row corresponds to the infinite bandgap. The shape of these vortex solitons should be a six-pole profile in which the centers of each pole are arranged symmetrically around a complete circle. However, as these solitons are coming from far from the edge of the bandgap ($\mu = 10.67$ for the first bandgap and $\mu = 10.75$ for the infinite bandgap), their profiles cannot form a complete circle. We will call this "incomplete". In detail, the number of the poles decreases as $\mu$ is far from the edge, which indicates that $\mu$ greatly affects the profiles of vortex solitons. Moreover, the profiles of the vortex solitons with larger $d$ tend to be incomplete.

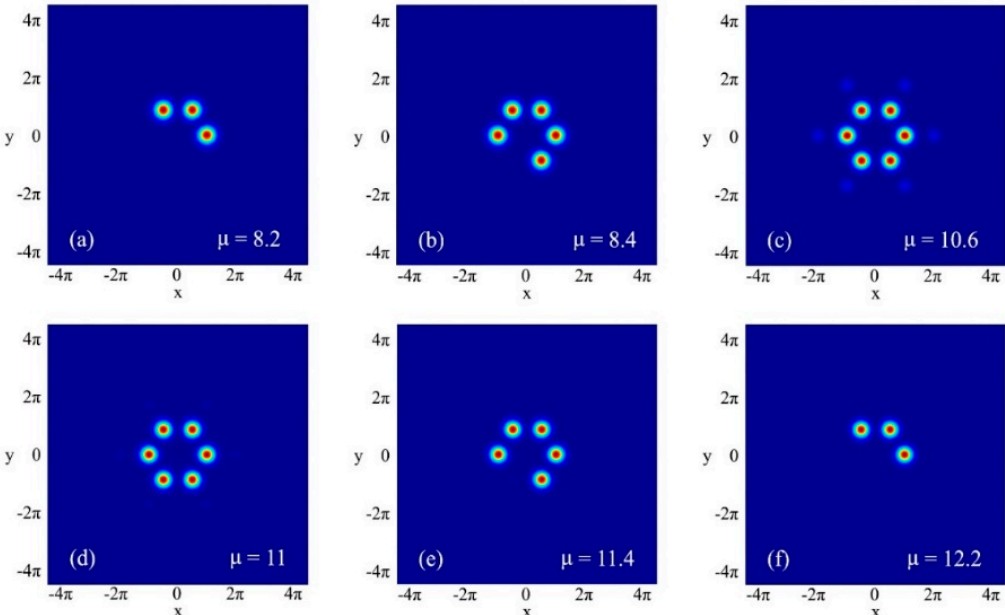

**Figure 12.** The evolution of the profiles of vortex solitons with $m = 3, d = 0.5$. (**a–f**) are the profiles of the vortex solitons with $\mu = 8.2, 8.4, 10.6, 11, 11.4, 12.2$, respectively. The upper row corresponds to solitons in the first bandgap, while the lower row corresponds to solitons in the infinite bandgap.

As for the vortex solitons with $d = 0.5, m = 2$, the evolution of their profiles is similar to that of the vortex solitons with $d = 0.5, m = 3$. To investigate the influence of the nonlocality, we consider the vortex solitons under strong nonlocal nonlinearity ($d = 2$) with $m = 2$, which are shown in Figure 13. For the vortex solitons near the edge of bandgap, additional humps are generated, as shown in Figure 13b,c. Their profiles turn into diamond shapes. When the vortex soliton is far from the edge of the bandgap, the profiles are complete (six poles arranged symmetrically around a circle) and then become incomplete, as shown in Figure 13a,d.

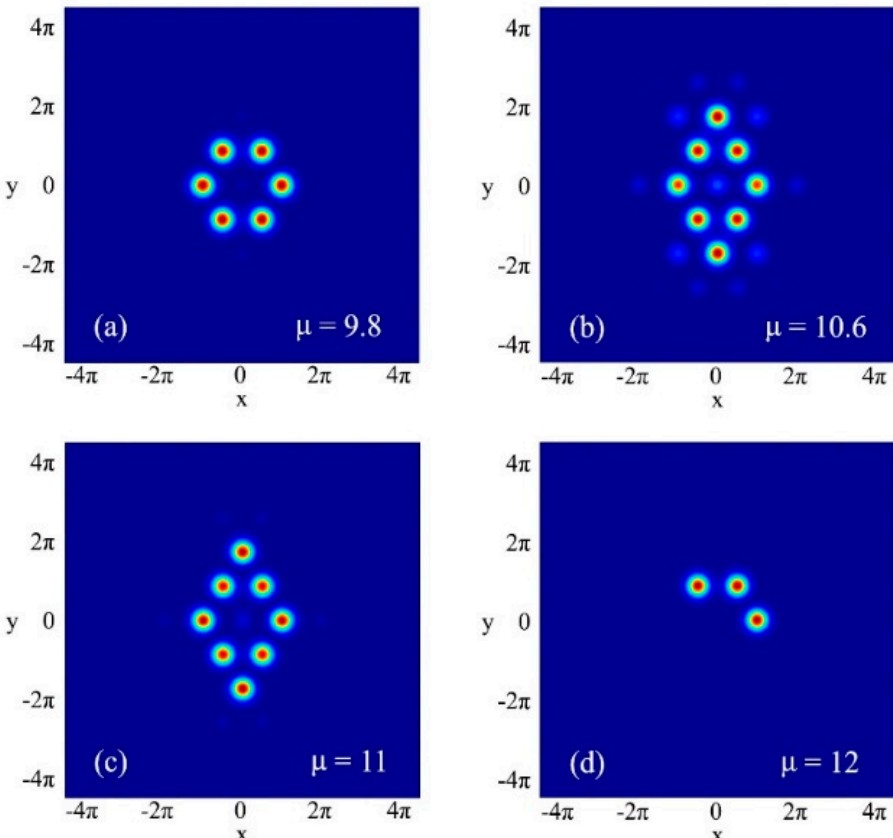

**Figure 13.** The evolution of the profiles of vortex solitons with $d = 2, m = 2$. (**a**–**d**) are the profiles of the vortex solitons with $\mu = 9.8, 10.6, 11, 12$, respectively. The upper row corresponds to solitons in the first bandgap, while the lower row corresponds to solitons in the infinite bandgap.

The evolution of these vortex solitons can be explained as follows. First, the strong nonlocality enhances the internal interaction of vortex solitons. The vortex solitons under stronger nonlocal nonlinearity have more potential to break the complete profile. Second, there are always linear modes inside the optical energy bands. Thus, solitons are harder to be localized as they are close to the edge of the bandgap. In the PT-symmetric triangular lattices, vortex solitons near the edge of the bandgap are likely to have more humps. Third, the main difference between the vortex solitons with $m = 2$ and $m = 3$ is the symmetry of their phase structures. From Figure 10, it is clear that the phase structure with $m = 3$ has higher symmetry than the phase structure with $m = 2$, which indicates the vortex solitons with $m = 3$ are more likely to maintain their profiles. The above analyses demonstrate that the propagation constant, the degree of nonlocality, and the topological charge codetermine the profiles of vortex solitons in nonlocal nonlinear triangular lattices.

## 4. Conclusions

Nonlocal dipole solitons, six-pole solitons, and vortex solitons can exist and propagate in the PT-symmetric triangular lattices with different gain/loss coefficients and different degrees of nonlocality.

However, not all solitons are linearly stable during propagation. For the dipole solitons, they can be stabilized when the degree of gain/loss is small enough. For the unstable six-pole solitons, there are two different situations. When its propagation constant is small, the power of this six-pole soliton is relatively large, which causes the collapse at the beginning of propagation. When the propagation constant is large, a new hotspot will appear in the center of this six-pole soliton due to the interaction of nonlocality and the refractive index. In addition, the existence range of these multipole solitons shows that multipole solitons are harder to be found as the nonlocality becomes stronger. For vortex solitons with different topological charges, we found that they exist in both the first bandgap and the infinite bandgap. Their profiles are codetermined by the propagation constant, degree of nonlocality, and topological charge. Although the profiles of these vortex solitons are different, all of them can maintain their initial profiles during their propagation; in other words, they are stable. These results reveal a feasible scheme for manipulating and stabilizing the beam in reconfigurable all-optical circuits and improving the bandwidth of information transmission.

**Author Contributions:** Conceptualization, Y.W. and J.H.; methodology, H.W.; software, Y.W. and P.C.; validation, H.W.; formal analysis, H.W.; investigation, Y.W. and J.H.; resources, H.W.; data curation, J.H. and Y.W.; writing—original draft preparation, J.H. and Y.W.; writing—review and editing, H.W.; visualization, Y.W.; supervision, H.W.; project administration, H.W.; funding acquisition, H.W.

**Funding:** This research was funded by the Science and Technologies plan Projects of Guangdong Province (Nos. 2017B010112003, 2017A050506013), Applied Technologies R & D Major Programs of Guangdong Province (Nos. 2015B010127013, 2016B01012300), Science and Technologies Projects of Guangzhou City (Nos. 201604046021, 201704030139, 201905010001), and Science and Technology Development Special Fund Projects of Zhongshan City (Nos. 2017F2FC0002, 2017A1009, 2019AG014).

**Conflicts of Interest:** The authors declare no conflict of interest.

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
