# Peer review of "Dynamics of Multipole Solitons and Vortex Solitons in PT-Symmetric Triangular Lattices with Nonlocal Nonlinearity"

_applsci, doi:10.3390/app9183731_

Round 1
Reviewer 1 Report
The paper should be published. The results are new and the topic is a nice addition to the various papers previously published by the authors.
Please consider this list of typos, English language corrections, and clarifications:
- line 17 and 222 and 229 and 232 and 237 and 238 and 322: “hotpots” —> “hotspots”
- line 28: “the diffraction balanced” —> “the diffraction is balanced”
- line 37: “it was firstly” —> “it was first”
- line 40-41: “PT symmetric system…” —> “A PT symmetric system…”
- line 46: “support complicate soliton” —> “support complicated solitons”
- line 48: “Many researches focus on” —> “Much research has focused on”
- line 50-51: The sentence “Thus, triangular lattices … stabilized.” could be improved perhaps to “Thus, triangular lattices are expected to exhibit new features of solitons upon propagation, such as the periodic beam oscillation inside a waveguide [24], and a certain parameter range where solitons can exist or be stabilized.”
- line 55: “still unexploited” —> “still unexplored”
- line 57: “Detailedly, we investigate the propagation” —> “We investigate in detail the propagation”
- line 60: “their stability are demonstrated” —> “their stability is demonstrated”
- line 61: “degrees of nonlocalities the gain-loss coefficient.” —> “degrees of nonlocalities in the gain-loss coefficient.”
- line 76-77: “None linear mode can exit in the “ —> “No linear mode can exist in the “
- line 96-97: “If the real part of $\lambda$ is zero, soliton is linearly stable.” —> “If the real part of $\lambda$ is zero, the soliton is linearly stable.”
- line 98: “, thus $\lambda$ is named perturbation growth rate” —> “, thus $\lambda$ is named the perturbation growth rate”
- line 104: “As showed” —> “As shown”
- line 129: “To detailedly investigate the dynamics” —> “To investigate in detail the dynamics”
- line 132: “soliton keep its profile” —> “soliton keeps its profile”
- line 150: “this dipole soliton decay” —> “this dipole soliton decays”
- line 152: “the right hump whose original postiion is $(\pi, 0)$ decay” —> “the right hump whose original postiion is $(\pi, 0)$ decays”
- line 153: “the power finally concentrate in the left hump” —> “the power finally concentrates in the left hump”
- line 154: “which indicate” —> “which indicates”
- line 161: “hump decay” —> “hump decays”
- line 164 “is larger the soliton” —> “is larger than the soliton”
- line 175: What is the meaning of “harder” in the sentence “harder multipole solitons can exist”?
- line 175-176: “as Fig 1 show” —> “as Fig 1 shows”
- line 179: “this areas” —> “these areas”
- line 206-207: “of which structure is a regular hexagon” —> “of which the structure is a regular hexagon”
- line 227: “eventually concentrate” —> “eventually concentrates”
- line 236: “is as follow” —> “is as follows”
- line 272: “The phase structures evolution” —> “The phase structures’ evolution”
- line 283-288: The definition of complete and incomplete is not clear the first few times this terminology is used. Perhaps consider rephrasing as follows (or similar). “The shape of these vortex solitons should be a six-pole profile in which the centers of each pole are arranged symmetrically around a complete circle. However, as these solitons are coming from far from the edge of the bandgap ($\mu = 10.67$ for the first bandgap and $\mu = 10.75$ for the infinite bandgap), their profiles cannot form a complete circle. We will call this ``incomplete”. In detail, the number of poles becomes less as $\mu$ is far from the edge, which indicates that $\mu$ affects the profiles of the vortex solitons greatly. Moreover, the profiles of the vortex solitons with larger $d$ tend to be incomplete in the sense which we describe above.”
- line 295: “showed in Fig. 13.” —> “shown in Fig. 13.”
- line 298: The usage of the word “normal” is not clear. Is this meant to be the same as “complete”? Perhaps this can be clarified by changing “the profiles become normal (six-pole)” to “the profiles are complete (six-pole arranged symmetrically around a circle)”
- line 321: “which cause the collapse” —> “which causes the collapse”
Additional Figure Caption Comments
- line 126: “thus that the white” —> “thus the white”
- line 167: “The dynamics of unstable dipole” —> “The dynamics of the unstable dipole”
- line 189: “can exists” —> “can exist”
- line 202: “The gray area in $\mu-W_0$. plane” —> “The gray area in the $\mu-W_0$ plane”
Other minor corrections:
- line 56: The indent for this paragraph is missing.
- after each displayed equation, the following text should not have indents since they are not a new paragraphs
- line 89: there is a box which appears after |F| and |G|, check the symbol there
- line 109: remove the word “Subsection”, just “3.1 Dipole solitons” would be better
Author Response
Dear Reviewer:
Thank you for reminding us of writing mistakes in the previous manuscript. The terms listed above were corrected in the revised manuscript. And we have checked up the whole manuscript carefully to avoid language errors. Our reply is as follows:
Review Report and our Reply:
The paper should be published. The results are new and the topic is a nice addition to the various papers previously published by the authors.
Please consider this list of typos, English language corrections, and clarifications:
Reply: Thank you for reminding us of writing mistakes in the previous manuscript. The terms listed were corrected in the revised manuscript, and we point out the location of correction after each term. We have checked up the whole manuscript carefully to avoid language errors.
- line 17 and 222 and 229 and 232 and 237 and 238 and 322: “hotpots” —> “hotspots”
Correction location: line 21, 235, 242, 245, 250, 251, 252 and 336.
- line 28: “the diffraction balanced” —> “the diffraction is balanced”
Correction location: line 31.
- line 37: “it was firstly” —> “it was first”
Correction location: line 40.
- line 40-41: “PT symmetric system…” —> “A PT symmetric system…”
Correction location: line 43.
- line 46: “support complicate soliton” —> “support complicated solitons”
Correction location: line 49.
- line 48: “Many researches focus on” —> “Much research has focused on”
Correction location: line 51.
- line 50-51: The sentence “Thus, triangular lattices … stabilized.” could be improved perhaps to “Thus, triangular lattices are expected to exhibit new features of solitons upon propagation, such as the periodic beam oscillation inside a waveguide [24], and a certain parameter range where solitons can exist or be stabilized.”
Correction location: line 53-56.
- line 55: “still unexploited” —> “still unexplored”
Correction location: line 58.
- line 57: “Detailedly, we investigate the propagation” —> “We investigate in detail the propagation”
Correction location: line 61.
- line 60: “their stability are demonstrated” —> “their stability is demonstrated”
Correction location: line 66.
- line 61: “degrees of nonlocalities the gain-loss coefficient.” —> “degrees of nonlocalities in the gain-loss coefficient.”
Correction location: line 67.
- line 76-77: “None linear mode can exit in the “ —> “No linear mode can exist in the “
Correction location: line 89-90.
- line 96-97: “If the real part of $\lambda$ is zero, soliton is linearly stable.” —> “If the real part of $\lambda$ is zero, the soliton is linearly stable.”
Correction location: line 109.
- line 98: “, thus $\lambda$ is named perturbation growth rate” —> “, thus $\lambda$ is named the perturbation growth rate”
Correction location: line 111.
- line 104: “As showed” —> “As shown”
Correction location: line 117.
- line 129: “To detailedly investigate the dynamics” —> “To investigate in detail the dynamics”
Correction location: line 142.
- line 132: “soliton keep its profile” —> “soliton keeps its profile”
Correction location: line 145.
- line 150: “this dipole soliton decay” —> “this dipole soliton decays”
Correction location: line 163.
- line 152: “the right hump whose original position is $(\pi, 0)$ decay” —> “the right hump whose original position is $(\pi, 0)$ decays”
Correction location: line 165.
- line 153: “the power finally concentrate in the left hump” —> “the power finally concentrates in the left hump”
Correction location: line 166.
- line 154: “which indicate” —> “which indicates”
Correction location: line 167.
- line 161: “hump decay” —> “hump decays”
Correction location: line 174.
- line 164 “is larger the soliton” —> “is larger than the soliton”
Correction location: line 177.
- line 175: What is the meaning of “harder” in the sentence “harder multipole solitons can exist”?
Correction location: line 188. The revised sentence is: “In general, less multipole soliton can exist in the first gap with larger degree of gain-loss W_0, as Fig. 2 show.”
- line 175-176: “as Fig 1 show” —> “as Fig 1 shows”
Correction location: line 189.
- line 179: “this areas” —> “these areas”
Correction location: line 192.
- line 206-207: “of which structure is a regular hexagon” —> “of which the structure is a regular hexagon”
Correction location: line 219.
- line 227: “eventually concentrate” —> “eventually concentrates”
Correction location: line 240.
- line 236: “is as follow” —> “is as follows”
Correction location: line 249.
- line 272: “The phase structures evolution” —> “The phase structures’ evolution”
Correction location: line 285.
- line 283-288: The definition of complete and incomplete is not clear the first few times this terminology is used. Perhaps consider rephrasing as follows (or similar). “The shape of these vortex solitons should be a six-pole profile in which the centers of each pole are arranged symmetrically around a complete circle. However, as these solitons are coming from far from the edge of the bandgap ($\mu = 10.67$ for the first bandgap and $\mu = 10.75$ for the infinite bandgap), their profiles cannot form a complete circle. We will call this ``incomplete”. In detail, the number of poles becomes less as $\mu$ is far from the edge, which indicates that $\mu$ affects the profiles of the vortex solitons greatly. Moreover, the profiles of the vortex solitons with larger $d$ tend to be incomplete in the sense which we describe above.”
Correction location: line 296-300.
- line 295: “showed in Fig. 13.” —> “shown in Fig. 13.”
Correction location: line 309.
- line 298: The usage of the word “normal” is not clear. Is this meant to be the same as “complete”? Perhaps this can be clarified by changing “the profiles become normal (six-pole)” to “the profiles are complete (six-pole arranged symmetrically around a circle)”
Correction location: line 312. Thanks for your suggest, and we agree this correction.
- line 321: “which cause the collapse” —> “which causes the collapse”
Correction location: line 335.
Additional Figure Caption Comments
- line 126: “thus that the white” —> “thus the white”
Correction location: line 139.
- line 167: “The dynamics of unstable dipole” —> “The dynamics of the unstable dipole”
Correction location: line 180.
- line 189: “can exists” —> “can exist”
Correction location: line 202.
- line 202: “The gray area in $\mu-W_0$. plane” —> “The gray area in the $\mu-W_0$ plane”
Correction location: line 215.
Other minor corrections:
- line 56: The indent for this paragraph is missing.
- after each displayed equation, the following text should not have indents since they are not a new paragraphs
- line 89: there is a box which appears after |F| and |G|, check the symbol there
- line 109: remove the word “Subsection”, just “3.1 Dipole solitons” would be better
Reply: The terms listed above were corrected. Thank you again for your suggestions about the format of our manuscript.
Yours sincerely
Hong Wang
Profrssor & Ph.D for Optoelectronics
Directcor, Engineering Research Center for Optoelectronics of Guangdong Province
School of Elecronics and Information Engineering
South China university of Technology

Reviewer 2 Report
The authors investigate the propagation characteristics of the multipole solitons and the evolution of the vortex solitons in triangular lattices. As main result they find that the dipole and the six-pole solitons can be found in the first bandgap of triangular lattices. The manuscript is interesting but in my opinion it is devoted to few specialists/experts of particular soliton propagation, it needs before publication main changes in order to have more impact also on other researchers in optics: 1) the introduction should better describe the state of the art, the practical applications in optics or the theoretical impact. It is strongly lacking from this point of view; 2) the theoretical model should be written after a detailed description of the problem/device/structure to be solved an the aim of the investigation, it seem written for mathematics researchers not for physicists/engineers; 3) the conclusion and the practical applications should be underlined since the name of the journal is “Applied Science”
Author Response
Dear Reviewer:
Thank you for pointing out what needs to be improved in this manuscript. Our reply is as follows:
Review Report and our Reply:
The authors investigate the propagation characteristics of the multipole solitons and the evolution of the vortex solitons in triangular lattices. As main result they find that the dipole and the six-pole solitons can be found in the first bandgap of triangular lattices. The manuscript is interesting but in my opinion it is devoted to few specialists/experts of particular soliton propagation, it needs before publication main changes in order to have more impact also on other researchers in optics:
1) the introduction should better describe the state of the art, the practical applications in optics or the theoretical impact. It is strongly lacking from this point of view;
Reply: We add the practical applications of our work in Introduction part. The corresponding sentences are:
Line 58-60: In practice, spatial optical solitons in nonlocal nonlinear and PT symmetric triangular lattices can be used for beam manipulation and all-optical switching, thus provide a scheme for creating reconfigurable all-optical circuits.
Line 63-64: These two types of soliton have complex phases and wave fronts, which means that they will have more complex dynamics and carry more information than fundamental solitons.
2) the theoretical model should be written after a detailed description of the problem/device/structure to be solved at the aim of the investigation, it seem written for mathematics researchers not for physicists/engineers;
Reply: We rewrite the description of the theoretical model in line 72-83. We describe each parameter in detail; thus, readers can understand the physical settings of this model.
3) the conclusion and the practical applications should be underlined since the name of the journal is “Applied Science”
Reply: We add the practical applications in Conclusion part. The corresponding sentence is:
Line 342-344: These results reveal a feasible scheme for manipulating and stabilizing the beam in reconfigurable all-optical circuits and improve the bandwidth of information transmission.
Thank you again for these helpful advices.
Yours sincerely
Hong Wang
Profrssor & Ph.D for Optoelectronics
Directcor, Engineering Research Center for Optoelectronics of Guangdong Province
School of Elecronics and Information Engineering
South China university of Technology

Round 2
Reviewer 2 Report
The paper can be now published since the auhors made a few changes but as wrote in the past revision I think the the part describing the physical problem must be properly improved (also now line 52-62 in my opinion are not sufficient and the manuscript could could be more interesting with a larger introduction and reference to practical implementation/applications)
Author Response
Dear Reviewer:
Thank you for pointing out what needs to be improved in this manuscript. Our reply is as follows:
Review Report and our Reply:
The paper can be now published since the authors made a few changes but as wrote in the past revision I think the part describing the physical problem must be properly improved (also now line 52-62 in my opinion are not sufficient and the manuscript could be more interesting with a larger introduction and reference to practical implementation/applications)
Reply: We rewrote the practical applications part in Introduction and cited relative references to enrich Introduction. The corresponding sentences are in line 58-62 now and are marked in blue.
Line 58-62: In practice, the dynamics and stability of spatial optical solitons and the interaction between them can be used in optical interconnect [27, 28], beam manipulation [29], optical computing [30], all-optical switching [31, 32] and logic gates [33-35]. With appropriate improvements, solitons in nonlocal nonlinear and PT symmetric triangular lattices may achieve the same function, thus provide a scheme for creating reconfigurable all-optical circuits.
[27] Friedrich L.; Stegeman G.I.; Millar P.; Aitchison J.S. 1 x 4 optical interconnect using electronically controlled angle steering of spatial solitons. IEEE Photonics Technology Letters 1999, 11, 988 - 990.
[28] Piccardi A.; Alberucci A.; Bortolozzo U.; Residori S.; Assant G. Readdressable Interconnects With Spatial Soliton Waveguides in Liquid Crystal Light Valves. IEEE Photonics Technology Letters 2010, 22, 694-696.
[29] Zhang H.; Xu D.; Zou J.; Zeng H.; Tian Y. Soliton control in inhomogeneous nonlocal media. Opt. Commun. 2011, 284, 1370-1378.
[30] Jakubowski M.H.; Steiglitz K.; Squier R. State transformations of colliding optical solitons and possible application to computation in bulk media. Phys. Rev. E 1998, 58, 6752-6758.
[31] Al-Khawaja U.; Al-Marzoug S.M.; Bahlouli H. All-optical switches, unidirectional flow, and logic gates with discrete solitons in waveguide arrays. Opt. Express 2016, 24, 11062-11074.
[32] Wu Y. New all-optical switch based on the spatial soliton repulsion. Opt. Express 2006, 14, 4005-4012.
[33] Ghadi A.; Sohrabfar S. All-Optical Multiple Logic Gates Based on Spatial Optical Soliton Interactions. IEEE Photon. Technol. Lett. 2018, 30, 569-572.
[34] Kheradmand R.; Aghdami K.M.; Khiaban B.H. Tunable all-optical gates in 2D discrete cavity solitons with local defect. Eur. Phys. J. D 2015, 69, 274.
[35] Scheuer J.; Orenstein M. All-optical gates facilitated by soliton interactions in a multilayered Kerr medium. J. Opt. Soc. Am. B 2005, 22, 1260-1267.
Thank you again for these helpful advices.
Yours sincerely
Hong Wang
Profrssor & Ph.D for Optoelectronics
Directcor, Engineering Research Center for Optoelectronics of Guangdong Province
School of Elecronics and Information Engineering
South China university of Technology
